# Thermal Effects of High-Intensity Laser Therapy on the Temporomandibular Joint Area in Clinically Healthy Racehorses—A Pilot Study

**DOI:** 10.3390/ani15101426

**Published:** 2025-05-15

**Authors:** Maria Soroko-Dubrovina, Paulina Zielińska, Krzysztof D. Dudek, Karolina Śniegucka, Karolina Nawrot

**Affiliations:** 1Institute of Animal Breeding, Wroclaw University of Environmental and Life Sciences, Chelmonskiego 38C, 51-160 Wroclaw, Poland; karolina.sniegucka@gmail.com (K.Ś.); karolina.nawrot@upwr.edu.pl (K.N.); 2Department of Surgery, Wroclaw University of Environmental and Life Sciences, pl. Grunwaldzki 51, 50-366 Wroclaw, Poland; paulina.zielinska@upwr.edu.pl; 3Faculty of Mechanical Engineering, Wroclaw University of Science and Technology, 50-370 Wroclaw, Poland; krzysztof.dudek@pwr.edu.pl

**Keywords:** temporomandibular joint, high-intensity laser therapy, infrared thermography, horses, safety

## Abstract

Temporomandibular joint (TMJ) disorders are a common issue in horses, often going unnoticed despite their potential impact on performance and well-being. High-intensity laser therapy (HILT) is a modern, non-invasive treatment that uses concentrated light energy to stimulate photomodulation, which may help in managing joint conditions. This study aimed to evaluate the effects of HILT on the TMJ area in clinically healthy racehorses. A total of 21 Thoroughbred horses underwent thermographic examinations before and after a single HILT session. The results showed a significant increase in body surface temperature at the treated joint, confirming the thermal effect of HILT. No adverse reactions or discomfort were observed in the horses, suggesting that this therapy is safe when applied with the appropriate settings. These findings contribute to the development of standardized laser therapy protocols for equine TMJ treatment. Future research should explore the long-term benefits and effectiveness of this treatment in horses with TMJ disorders.

## 1. Introduction

High-intensity laser therapy (HILT) has recently been introduced in the field of physical therapy in veterinary medicine [1]. It is a non-invasive therapy that utilizes concentrated light generated by a class IV laser device with a power exceeding 0.5 W [2]. The light employed spans the visible to infrared spectrum and can be delivered in either continuous or pulsed modes, resulting in the dispersion of laser energy within the irradiated tissue [3].

One of the preconditions of improving HILT procedures in equine veterinary medicine is the determination of the thermal effect, which is a result of the transformation of absorbed light energy to heat [4]. The thermal effect in tissues is governed by the absorption properties of key endogenous molecules including water, hemoglobin, and melanin. Each of these chromophores interacts with light in distinct ways, absorbing energy at specific wavelengths. For instance, water exhibits strong absorption in the infrared spectrum, whereas hemoglobin and melanin are more responsive to light in the visible and near-infrared ranges. This selective absorption is a critical factor in determining tissue response to thermal treatments. Consequently, the choice of laser wavelength must be carefully tailored to the target chromophore to achieve the desired therapeutic effect. The wavelength not only influences the depth of light penetration but also shapes the thermal dynamics within the tissue, underscoring its importance in medical applications [5].

The thermal effect induces changes in blood flow, enhances the permeability of blood vessels, and accelerates cellular metabolic responses. It also stimulates collagen production within soft tissue, increasing blood flow and vascular permeability. As such, the heat generated during HILT–tissue interaction can be utilized therapeutically and for regeneration through controlled heat delivery [6]. However, thermal effects also raise safety concerns when unintended interaction pathways occur in medical applications [7]. Excessive laser energy or prolonged exposure can elevate temperatures to damaging levels, causing burns or injuries [8]. To mitigate these risks, the correct selection of laser energy power, light wavelength, and pulse duration is critical in determining the specific interaction effects.

The first studies dealing with the application of HILT in equine veterinary medicine used thermal effects to treat tendinopathy and desmopathy [9,10,11], kissing spines syndrome [12], and osteoarthritis [13]. Recent studies have focused on the thermal effectiveness of HILT in the distal limbs of clinically healthy horses. One study conducted laser therapy on the superficial digital flexor tendon, finding that body surface temperature increased by an average of 3.5 °C after HILT [14]. Other studies have evaluated the thermal effects of HILT on the tarsal and fetlock joints. The temperature at the tarsal joint was found to significantly increase by 2.5 °C after HILT [15], whereas the temperature at the fetlock joint area temperature increased by an average of 3 °C after HILT [16]. Additional research is required to determine the thermal effect of HILT on different types of joints, including regular treatment, in order to more accurately determine the correct and safe laser parameters. Diseases of the temporomandibular joint (TMJ) are a common problem in sport horses. Over 35% of horses older than 1 year of age have been reported to have osteoarthritic changes in the TMJ, without any clinical signs [17]. While certain changes in the TMJ may not initially manifest as obvious clinical signs, research indicates that TMJ dysfunctions can still affect performance and behavior in horses. TMJ disorders in horses can present with subtle symptoms such as resistance to the bit, difficulty chewing, head tossing, or overall poor performance, even in the absence of overt lameness or pain [18]. A study presented by Carmalt et al. [19] demonstrated that acute TMJ inflammation in horses significantly affected rein tension and movement symmetry, suggesting that even minor or early-stage pathology may alter biomechanics and athletic function.

TMJ is a uniquely specialized, bilaterally functioning joint characterized by its fibrocartilaginous articular surfaces and dense sensory innervation, primarily from the mandibular branch of the trigeminal nerve (CN V3). As such, it is particularly sensitive to mechanical stress [20]. It is richly connected to the branches of the maxillary artery, including the temporomandibular articular branches and the masseteric artery, which provide essential blood flow to the joint structures [21]. Due to its complex anatomy and functional significance, TMJ represents a suitable target for the application of localized therapeutic approaches. HILT has demonstrated efficacy in reducing pain and improving joint function in human patients with TMJ disorders, making it a promising treatment modality [22]. As such, the objective of the study was to assess the thermal effects of HILT on the TMJ area of clinically healthy Thoroughbred racehorses. It was hypothesized that HILT would increase body surface temperature at the TMJ area and that it is a safe therapy for the TMJ area in healthy tissue.

## 2. Materials and Methods

The Animal Welfare Advisory Team at the Wroclaw University of Environmental and Life Sciences approved the study design, in compliance with Polish and European Union legislation on animal experimentation (no. 6/2025). The procedures used in this study were deemed not to cause pain, suffering, distress or lasting harm equivalent to or greater than that caused by the introduction of a needle (Article 1.5 of EU Directive 2010/63/EU).

### 2.1. Study Population and Data Collection

A blind, randomized, sham-controlled study was conducted on 21 clinically healthy Thoroughbreds aged 2–5 years old. The horses were in regular racing training at the Partynice Racecourse in Wroclaw (Poland). They were housed in individual boxes (3.5 × 3.5 m) within the same stable, with a common management and training regime. All the horses were clinically healthy without any signs of inflammation or other issues in the head area, and they did not present any stereotypical behavior. Skin pigmentation in the treatment area was assessed as pigmented for all horses.

The horses were fed three times a day at 05.30 a.m., 12.30 p.m., and 6.30 p.m. The diets, which were prepared individually for each horse, met the Nutrient Requirements of Horses [23] and included meadow hay, concentrate mixture (oat grain with muesli “Livery Mix”, Saracen, United Kingdom), and vegetable oil (0.5 mL/kg of body weight).

On the day of the experiment, each horse was subjected to a thermographic examination in order to determine any changes in the body surface temperature within the area of the TMJ. Thermographic images of the head were taken from the left and right sides. The left TMJ was subjected to HILT, while the right TMJ was the control area (without HILT application). The images of the left and right sides of the head were taken just before and immediately after HILT application. The average surface temperature was measured in reference to both the right and left TMJ.

### 2.2. Thermographic Examination

Thermal imaging was conducted using a Vario Cam hr resolution infrared camera (un-cooled microbolometer focal plane array with a sensor size of 640 × 480, spectral range 7.5–14 μm, noise equivalent temperature difference of <20 mK at 30 °C, using the normal lens with an IFOV of 0.57 mrad, measurement uncertainty of ±1% of the overall temperature range, InfraTec, Dresden, Germany). The thermographic examination followed the same protocol as described in our previous research [15], aiming to reduce the influence of environmental factors like air draughts and sunlight. Therefore, all imaging was performed in a closed stable environment, inside the horse’s box, at a controlled ambient temperature of approximately 22 °C, as measured by a TES 1314 thermometer (TES, Taipei, Taiwan). The horses under examination were at rest—before their daily exercises—and were brushed 1 h before the examination [16,24]. The same operator (M.S.-D.) conducted all the thermographic measurements to ensure consistency. The horses were restrained by one person in the box with a headcollar. The distance between the camera and the horse was standardized at 0.5 m, at an angle of 90 degrees to the head, and the emissivity was uniformly set to 1 for all the assessments. The mean body surface temperature was calculated over a defined round region marked in the TMJ area (Figure 1), using IRBIS 3 Professional software (InfraTec, Dresden, Germany).

### 2.3. High-Intensity Laser Therapy

HILT was performed using a Polaris HP S class 4 laser (Astar, Bielsko-Biała, Poland). The following two infrared wavelengths with different parameters were used: 808 nm (25 J/cm^2^, 1.5 W, 10 Hz, time of treatment 73 s, total energy delivered 100 J) and 980 nm (20 J/cm^2^, 1.5 W, 2000 Hz, time of treatment 66 s, total energy 80 J). Both wavelengths were delivered simultaneously. The treatment area (4 cm^2^) was unshaven. The horses had a natural coat, with an average hair length of approximately 0.5 cm. There was no other skin preparation performed before the HILT. A contact and labile method of the therapy using slow circular movements of the probe over the joint area was performed. The diameter of the radiating area of the laser handpiece used in this study was 1 cm^2^. The distance from the source of the laser light to the skin was 1 cm. The same person performed the procedure (P.Z.) while wearing safety glasses. The horse’s eyes were covered by hand during the therapy, by the person holding the horse.

### 2.4. Statistical Analysis

A statistical analysis of the results was conducted using Statistica v.13.3 (TIBCO Software Inc., Palo Alto, CA, USA). The empirical distributions of the measured temperatures in both joints before and after HILT did not significantly deviate from a normal distribution, as verified by the Shapiro–Wilk test. Homogeneity of variances was assessed using the Brown–Forsythe test and Levene’s test, while the significance of differences within samples was evaluated using the *t*-test for paired samples. A statistical test result was considered significant when the two-tailed probability was *p* < 0.05.

## 3. Results

The average body surface temperature in the TMJ area increased significantly after HILT (*p* < 0.001). The body surface temperature of the area examined was higher (by a mean of 2.02 °C) after HILT, compared to the temperature before HILT (Table 1). The temperature change was significantly different from zero (*p* < 0.001). The body surface temperature of the control TMJ did not change significantly.

## 4. Discussion

This study has demonstrated the thermal effect of HILT when applied to a healthy TMJ joint. The body surface temperature at the joint increased by an average of 2.0 °C, supporting the study hypothesis. The thermal effect of HILT was also confirmed in our previous research, conducted on clinically healthy joints in the distal parts of the limbs. The body surface temperature of the fetlock joint was found to increase significantly by an average of 3 °C just after therapy [16], while for the tarsal joint, an increase of 2.53 °C was reported [15]. Significant results were also found in another of our previous studies, where we identified differences in the effects of HILT on pigmented vs. non-pigmented skin at a clinically healthy fetlock joint. In horses with pigmented skin, the body surface temperature increased, whereas in the non-pigmented skin group, the temperature decreased [24]. Therefore, in this study, only horses with pigmented skin were included to ensure consistency in light absorption and thermal response.

HILT is increasingly being utilized in the treatment of TMJ disorders in humans, particularly for managing pain, chewing dysfunctions, and chronic inflammation. Clinical studies have demonstrated that HILT effectively reduces pain and improves TMJ mobility through mechanisms related to the thermal effect, increased microcirculation, and reduced muscle tension in the joint region [25]. Furthermore, treatment of the TMJ region may exert a neuroprotective or neuromodulatory effect on the trigeminal nerve (CN V3), potentially contributing to a reduction in the neuropathic pain associated with TMJ disorders [26].

Treating TMJ disorders with therapeutic devices presents challenges due to their complex anatomy, delicate structure, and proximity to critical components such as nerves and blood vessels, making treatment particularly challenging [27]. The thermal effects of HILT are closely linked to laser safety concerns. Excessive laser energy or prolonged exposure can lead to significant temperature increases, potentially causing tissue damage, burns, or other injuries. To minimize these risks, strict safety measures must be implemented, covering both laser system parameters and personnel training. Adhering to the established safety standards, such as those outlined in the international guidelines, ensures that laser therapy delivers its intended benefits while maintaining a high level of safety [28]. According to Hinchcliff et al. [29], irradiation near the eyes is considered a contraindication in equine practice due to the potential for retinal damage. In the present study, the appropriate safety measures were implemented to mitigate this risk. The horse’s eyes were covered by hand throughout the procedure, with the beam carefully oriented away from the orbital area. These precautions are consistent with safety recommendations described in the human clinical literature, where eye protection is strongly emphasized during laser procedures in the craniofacial region [30,31].

TMJ laser therapy is particularly liable to overheating tissue, due to the small treatment area and extremely limited lability of the therapy. The HILT treatment parameters applied in the present study did not result in any clinical abnormalities or adverse effects in the irradiated joint. Furthermore, no changes were observed in the horses’ behavior, indicating that the therapy did not cause discomfort or stress. These findings suggest that the HILT parameters used in this study are safe and well tolerated by horses. The absence of negative reactions reinforces the potential of HILT as a viable treatment modality for TMJ dysfunctions.

TMJ disorders in horses encompass conditions such as osteoarthritis, trauma-related fractures, and septic arthritis, which may lead to pain, behavioral issues, or reduced performance, even in the absence of overt clinical signs [18,19]. While traditional therapies include systemic anti-inflammatories or intra-articular injections, their long-term effectiveness remains uncertain, and more advanced, non-invasive treatment options are needed. HILT has shown promising results in human TMJ therapy, particularly in reducing inflammation, improving joint function, and modulating neuromuscular pain [22,25]. Due to its thermal effect, HILT may offer a valuable adjunct or alternative in the management of equine TMJ pathologies. Further research on horses with diagnosed TMJ disorders is necessary to evaluate the therapy’s clinical effectiveness.

Despite the significant findings of this study, certain limitations should be acknowledged. Firstly, this study was a pilot study with a relatively small sample size (n = 21), limiting the generalizability of the findings to a broader population of racehorses. Future research should include larger study groups and horses with different skin pigmentation levels to determine whether pigmentation influences laser energy absorption or thermal effects. The second major limitation was the application of only one set of HILT parameters without comparing different therapeutic settings. As standardized protocols for HILT in TMJ treatment have not yet been established, future studies should compare different laser wavelengths, energy densities, and application modes to determine the most effective treatment parameters.

The third limitation concerns the absence of a sham group to evaluate whether the mere contact of the laser device with the skin contributed to the temperature increase. Even in contact-based techniques without laser emission, heat may be generated through thermal conduction or mechanical friction, potentially leading to temperature increases independent of the thermal effect [32].

Used in the methodology, thermography is widely recognized as a reliable and repeatable method for assessing surface temperature changes in horses. In a study by Tunley and Henson [33], thermographic patterns in the thoracolumbar region remained consistent over a 7-day period, demonstrating the method’s reproducibility without the need for equilibration time. Additionally, studies by Na Lampang et al. [34] and Roy et al. [35] highlighted that, under controlled environmental conditions, thermography provided consistent temperature readings across various anatomical regions, further reinforcing its reliability for clinical assessments.

## 5. Conclusions

Our findings confirm the thermal effects of HILT on the TMJ, contributing to the establishment of appropriate and safe HILT parameters for the treatment of TMJ injuries and inflammation. There is a need for further research to standardize treatment protocols and determine the short- and long-term effectiveness of laser therapy in the treatment of TMJ disorders.

## Figures and Tables

**Figure 1 animals-15-01426-f001:**
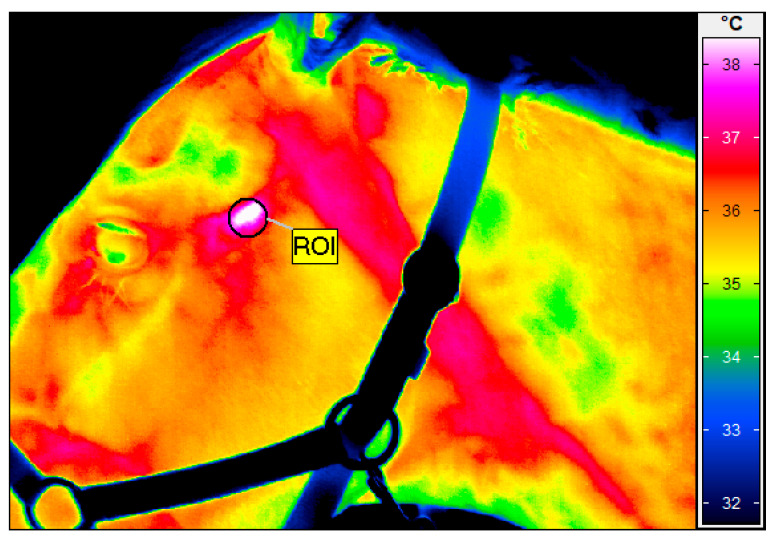
Thermographic image of left temporomandibular joint (TMJ) just after high-intensity laser therapy. Region of interest (ROI) indicates area of TMJ joint where average temperature was 39.4 °C.

**Table 1 animals-15-01426-t001:** Basic descriptive statistics of temporomandibular joint (TMJ) body surface temperature on left (with application of high-intensity laser therapy—HILT) and right (control) side before and after HILT.

TMJ	Mean (SD) (°C)	Result of the Test
Left joint, before HILT therapy	36.08 (0.56)	t = 1.948 *p* = 0.066
Right joint, before HILT therapy	36.01 (0.50)	
Left joint, after HILT therapy	38.10 (0.73)	t = 12.894 *p* < 0.001
Right joint, after HILT therapy	35.93 (0.48)	
TMJ left joint	2.02 (0.78)	t = 12.246 *p* < 0.001
TMJ right joint (control)	−0.08 (0.12)	

## Data Availability

The data that support the findings of this study are available from the corresponding author [M.S.-D.], upon reasonable request.

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
