# Peer review of "Thermal Effects of High-Intensity Laser Therapy on the Temporomandibular Joint Area in Clinically Healthy Racehorses—A Pilot Study"

_animals, 2025, doi:10.3390/ani15101426_

Round 1
Reviewer 1 Report
Comments and Suggestions for Authors
Abstract
Line 16: generating heat is not the therapy goal of HILT. Temperature increase is a by-product of photomodulation. Please adapt this sentence.
Line 25: consider to use "thermal effects" instead of '"phototermal effect" when you describe the temperature measured on the skin.
Introduction
Line 44-46: Please remove the sentence "It employs high peak....150 ms in duration". This relates to a specific setting of a specific laser device, and shouldn't be generalized for HILT.
Line 54: ...at specific wavelengths. Please add a reference.
Line 56: ...near-infrared ranges. Please add a reference.
Line 69: ...burns or injuries. Please add a reference.
Line 74: syndrome instead of syn-drome.
Line 84-86: If changes in the TMJ do not cause clinical signs, then why is there a need for therapy? Please add references pointing out why this pathology is relevant in horses.
Line 93-94: combination of an intricate.... improve joint function: Please make clear why HILT is suitable for TMJ treatment. The way it's described now makes it sound like that's only based on the anatomy of joint.
Material and Methods
Line100: Union instead of Un-ion.
Line 116: Was thermographic examination performed only once? Or were measurements repeated in order to test repeatability?
Line 129: ....same as in previous research. Please adapt in a way that readers can understand the procedure without have to read reference [15]. Maybe just make a summary of the procedure and refer to [15] thereafter.
Line 150: Why were horses unclipped? Please mention the length of the coat, as laser therapy has been described before mainly on a clipped skin.
Line 154: What's the diameter of the radiating area of the handpiece? What's the distance to the skin?
Results
Table 1: Please move Celcius to the correct row (Mean (SD)).
Line 183: Why were only horses with pigmented skin selected?
Line 184-192: I believe this section fits better in the introduction.
Line 190:...cartilage regeneration. Please add a reference.
Reference nr 21 is not included in the text, please remove it from the reference list.
Line 208: .....are not only effective but also.... Please remove. In this study only increase in skin temperature was measured, which cannot directly be related to effectiveness of this therapy. There were no effects of treatment analysed in this study.
Please add a safety concern of laser treatment close to the eye, which has been described as a contra-indication (Hinchcliff, Equine sports medicine and surgery) and point out clearly which safetly measures must be taken before treating the TMJ area (and maybe add some human studies on this topic).
Line 225: Please add someting about the repeatability / reliability of the chosen measure thermography.
Conclussion
Line 230: not only long term, please add short term.
Overall:
-Thermal effects of HILT on horse skin have been described before (reference 12 and 14), which makes this study less unique. In my opinion, results of this study are only relevant to determine safety for treatment of the TMJ. As such, this manuscript should focus a bit more TMJ pathologies and why HILT could be interesting as a treatment option. Please desribe these TMJ pathologies briefly, point out the need for treatment options of TMJ disorders in the horse, and discuss therapies available for TMJ disorders, prognosis etc.
-Also, be carefull claiming to have (phototermal) effects, since increase in temperature doens't neccessarily mean there is a phototermal effect (which includes cellular reactions, which cannot be analysed with this study setup).
-Please point out clearly the safety measures to avoid radiating the eye while treating the
TMJ.
-Please describe a bit more the thermography used to determine the termal effects, and describe the reliabilty of such a method based on literature in the horse.
Author Response
Responses to Reviewer 1 comments
Manuscript ID: animals-3582523, entitled “Thermal effects of high-intensity laser therapy on the temporomandibular joint area in clinically healthy racehorses – pilot study”
Comment: Line 16: generating heat is not the therapy goal of HILT. Temperature increase is a by-product of photomodulation. Please adapt this sentence.
Response: Thank you for your comment. The sentence has been corrected, Line:16.
Comment: Line 25: consider to use "thermal effects" instead of '"phototermal effect" when you describe the temperature measured on the skin.
Response: The word "phototermal effect has been changed to "thermal effects" in the whole text.
Comment: Line 44-46: Please remove the sentence "It employs high peak....150 ms in duration". This relates to a specific setting of a specific laser device, and shouldn't be generalized for HILT.
Response: The sentence has been removed, Line 45.
Comment: L Line 54: ...at specific wavelengths. Please add a reference.
Response: Thank you for your suggestion. The reference supporting this statement have been included at the end of the sentences to which they apply Line: 59.
Comment: L Line 56: ...near-infrared ranges. Please add a reference.
Response: The same comment as above.
Comment: Line 69: ...burns or injuries. Please add a reference.
Response: There reference has been added , Line 67.
Comment: Line 74: syndrome instead of syn-drome.
Response: The mistake has been corrected, Line 73.
Comment: Line 84-86: If changes in the TMJ do not cause clinical signs, then why is there a need for therapy? Please add references pointing out why this pathology is relevant in horses.
Response: Thank you for your comment. New text and reference have been added, Lines: 84-91.
Comment: Line 93-94: combination of an intricate.... improve joint function: Please make clear why HILT is suitable for TMJ treatment. The way it's described now makes it sound like that's only based on the anatomy of joint.
Response: Thank you for comment: Due to its complex anatomy and functional significance, TMJ represents a suitable target for the application of localized therapeutic approaches. HILT has demonstrated efficacy in reducing pain and improving joint function in patients with TMJ disorders, making it a promising and non-invasive treatment modality (Ekici et al. 2022). New text has been added, Lines: 97-101.
Comment: Line100: Union instead of Un-ion.
Response: Corrected :Line:107.
Comment: Line 116: Was thermographic examination performed only once? Or were measurements repeated in order to test repeatability?
Response: The thermographic examination performed only once.
Comment: Line 129: ....same as in previous research. Please adapt in a way that readers can understand the procedure without have to read reference [15]. Maybe just make a summary of the procedure and refer to [15] thereafter.
Response: Thank you for your comment. The text has been corrected, Lines: 135-140.
Comment: Line 150: Why were horses unclipped? Please mention the length of the coat, as laser therapy has been described before mainly on a clipped skin.
Response: Thank you for your valuable comment. The horses included in this study were not clipped, and this decision was based on findings from our previous research: Zielińska, P.; Soroko-Dubrovina, M.; Śniegucka, K.; Dudek, K.; Čebulj-Kadunc, N. (2023). Effects of High-Intensity Laser Therapy (HILT) on Skin Surface Temperature and Vein Diameter in Healthy Racehorses with Clipped and Non-Clipped Coat. Animals, 13(2), 216. This study demonstrated a significantly higher increase in body surface temperature in horses with a non-clipped coat compared to those with a clipped coat. These findings suggest that the hair coat plays a key role in light energy absorption and conversion into heat, supporting the thermal effect. The sentence “The horses had a natural, coat with an average hair length of approximately 0,5 cm” has been added in Line: 62-164.
Comment: Line 154: What's the diameter of the radiating area of the handpiece? What's the distance to the skin?
Response: The diameter of the radiating area of the laser handpiece used in this study was 1cm2. The distance from the source of the laser light to the skin was 1 cm. This information has been added, Line: 160-162.
Comment: Table 1: Please move Celcius to the correct row (Mean (SD)).
Response: Corrected.
Comment: Line 183: Why were only horses with pigmented skin selected?
Response: Thank you for your question. In the present study, only horses with pigmented skin were selected in order to ensure uniform light absorption characteristics across the treatment group. New text has been added, Lines: 193-194.
Comment: Line 184-192: I believe this section fits better in the introduction.
Response: Thank you for your valuable suggestion. We agree that the information regarding the application of HILT in human TMJ therapy is highly relevant. However, we chose to include this content in the Discussion section rather than the Introduction to better contextualize and compare our equine findings with existing clinical applications in humans.
Comment: Line 190:...cartilage regeneration. Please add a reference.
Response: This part of the text has been removed from the text.
Comment: Reference nr 21 is not included in the text, please remove it from the reference list.
Response: The reference nr 21 has been removed from the reference list
Comment: Line 208: .....are not only effective but also.... Please remove. In this study only increase in skin temperature was measured, which cannot directly be related to effectiveness of this therapy. There were no effects of treatment analysed in this study.
Response: Thank you for that comment. are not only effective but also – has been removed from the text.
Comment: Please add a safety concern of laser treatment close to the eye, which has been described as a contra-indication (Hinchcliff, Equine sports medicine and surgery) and point out clearly which safety measures must be taken before treating the TMJ area (and maybe add some human studies on this topic).
Response: The text has been added , Line: 211-218.
Comment: Line 225: Please add something about the repeatability / reliability of the chosen measure thermography.
Response: Thank you for that comment, new text has been added, Line:252-259.
Comment: Line 230: not only long term, please add short term.
Response: short term has been added, Line 269.
Comment: -Thermal effects of HILT on horse skin have been described before (reference 12 and 14), which makes this study less unique. In my opinion, results of this study are only relevant to determine safety for treatment of the TMJ. As such, this manuscript should focus a bit more TMJ pathologies and why HILT could be interesting as a treatment option. Please desribe these TMJ pathologies briefly, point out the need for treatment options of TMJ disorders in the horse, and discuss therapies available for TMJ disorders, prognosis etc.
Response: New text has been added, Lines: 227-236.
Comment: -Also, be carefull claiming to have (phototermal) effects, since increase in temperature doens't neccessarily mean there is a phototermal effect (which includes cellular reactions, which cannot be analysed with this study setup).
Response: Word photothermal has been changed to thermal effects in the whole text.
Comment: Please point out clearly the safety measures to avoid radiating the eye while treating the TMJ
Response: Thank you for your comment. We agree that proper safety measures are critical when applying HILT near eyes. New text has been added, Line: 211-218
Comment: -Please describe a bit more the thermography used to determine the termal effects, and describe the reliabilty of such a method based on literature in the horse.
Response: New parahraph has been added Line: 252-259.
Reviewer 2 Report
Comments and Suggestions for Authors
This pilot study fulfills the aim of ascertaining it high intensity laser (HILT) applied over healthy horse TMJs has a thermal effect.
As laser is becoming a commonly used modality in equine rehabilitation, and TMJ dysfunction is becoming increasingly recognised as a cause of equine poor performance, it is relevant to examine the effect of HILT on the region. There are some weaknesses
- It is a pilot study thus low numbers (recognised by authors)
- the design could have included a sham group (no contact of the laser with skin) (Recognised by authors)
- it is assumed that in increase in temperature measured with thermograph has a therapeutic benefit - the population studied was clinically healthy
- there is over-citation of the same authors
The strengths of this study is that it is the methodology is well described and the statistical anaylsis is simple yet appropriate. It is an area that will interest equine rehabilitation practitioners as laser is widely utilised.
A few typographical errors and awkward grammar
line 191 - how does the TMJ exert a neuroprotective effect on the trigeminal nerve - explain this in better format please - or do you mean treatment of the TMJ may do the same?
line 194 - what is a physical device?
There are a number of instances when words with two syllables are separated by a hyphen
- lines 198, 202, 208, 215
Author Response
Responses to Reviewer 2 comments
Manuscript ID: animals-3582523, entitled “Thermal effects of high-intensity laser therapy on the temporomandibular joint area in clinically healthy racehorses – pilot study”
Comment 1: This pilot study fulfills the aim of ascertaining it high intensity laser (HILT) applied over healthy horse TMJs has a thermal effect.
Response: Thank you for your comment. Indeed, the primary objective of this pilot study was to evaluate whether the application of HILT over the TMJ area in clinically healthy horses induces a measurable thermal effect.
Comment 2: As laser is becoming a commonly used modality in equine rehabilitation, and TMJ dysfunction is becoming increasingly recognised as a cause of equine poor performance, it is relevant to examine the effect of HILT on the region. There are some weaknesses
- It is a pilot study thus low numbers (recognised by authors)
- the design could have included a sham group (no contact of the laser with skin) (Recognised by authors)
- it is assumed that in increase in temperature measured with thermograph has a therapeutic benefit - the population studied was clinically healthy
- there is over-citation of the same authors
Response: 1–2. We appreciate your confirmation of the limitations that we have already acknowledged in the manuscript. As this was a pilot study, the number of horses was intentionally limited to assess feasibility and gather preliminary data. Similarly, we agree that the inclusion of a sham group (non-contact laser) would strengthen the study design, and we noted this limitation in the discussion. This will certainly be taken into account in future research planning.
- Thank you for this comment. As this is the first study investigating the HILT on the TMJ area in horses, we deliberately chose clinically healthy horses to establish a baseline and ensure safety before extending the study to horses with TMJ dysfunction. The results of this pilot study provide a basis for further research into the use of thermography for monitoring both short- and long-term changes and effects induced by HILT. This will be helpful in determining the most effective and optimal treatment parameters. Importantly, our findings indicate that HILT is safe for application in the TMJ region, which supports the rationale for conducting future studies on clinical cases. New text has been added Line 211-218; 227-125.
- We agree that there is a high rate of self-citation in the current version of the manuscript. We have reduced the number of self-citations in the manuscript.
Comment 3: The strengths of this study is that it is the methodology is well described and the statistical anaylsis is simple yet appropriate. It is an area that will interest equine rehabilitation practitioners as laser is widely utilised.
Response: Thank you very much for your positive feedback.
Comment 4: A few typographical errors and awkward grammar
Response: All typographical errors have been corrected in the text.
Comment 5: line 191 - how does the TMJ exert a neuroprotective effect on the trigeminal nerve - explain this in better format please - or do you mean treatment of the TMJ may do the same?
Response: Thank you for your comment. We have revised the sentence in the manuscript for: "Furthermore, treatment of the TMJ region may exert a neuroprotective or neuromodulatory effect on the trigeminal nerve (CN V3), potentially contributing to the reduction of neuropathic pain associated with TMJ disorders [25]” Lines 199-202
Comment 6: line 194 - what is a physical device?
Response: Thank you for your comment. By "physical devices," we refer to therapeutic equipment used in physical medicine and rehabilitation, such as high-intensity laser therapy devices, radial and focused shockwave therapy equipment etc. To avoid confusion, we changed physical device to therapeutic devices Line: 203.
Comment 7: There are a number of instances when words with two syllables are separated by a hyphen - lines 198, 202, 208, 215
Response: All mistakes have been corrected.
Round 2
Reviewer 1 Report
Comments and Suggestions for Authors
Thank you for your adaptations. I believe the manuscript has greatly improved and is suitable for publication.